# Ferdynand Ruszczyc: A Polish Painter at the Crossroads of Cultures

## Agnieszka Rosales Rodríguez

Institute of Art History, University of Warsaw, 00-927 Warszawa, Poland; a.rosales@uw.edu.pl

**Abstract:** The oeuvre of beloved Polish painter Ferdynand Ruszczyc (1870–1936) reflected the patriotic Neo-Romantic landscape trend of the fin-de-siècle prevalent in Germany and the Nordic countries (Denmark, Finland, Norway, Sweden). It should be considered in the context of Nordic visual culture for two reasons: (1) until the affiliation of Central and Eastern European nations with the Soviet Union in the wake of World War Two, nations bordering the Baltic formed a single, fluid territory of cultural exchange, and (2) Ruszczyc's oeuvre displays significant commonalities with dominant patriotic and Neo-Romantic trends of progressive artists around the Baltic Sea, where landscape became a vehicle for expressing dreams and emotions, as well as love of homeland. This article situates Ruszczyc's national and artistic identity at the crossroads of cultures and artistic impulses, regional as well as international. Ruszczyc was born in Bohdanów near Vilnius (now Belarus) to a Polish father and a Danish mother. Like many Polish artists from the Russian partition, he was educated at the Imperial Academy of Fine Arts in St. Petersburg, where he studied with Ivan Shishkin (1832–1898) and Arkhip Kuindzhi (1878–1910). He also travelled to Sweden. Ruszczyc was influenced by the Russian art circle *Mir Iskusstva* (World of Art, est. 1898) and is often compared with Nordic (e.g., Akseli Gallen-Kallela; Finnish, 1865–1931) and German (e.g., Otto Modersohn; 1865–1943) artists. His visions of nature are sometimes raw monumental images of the northern landscape or fairy-tale fantasies containing symbolic allusiveness and a mythical, poetic element that evoke intimate memories of the land of his childhood. In his paintings, Ruszczyc presented the changeability of seasons, orchards, soil and streams, clouds formations, and tree trunks with palpable emotion. By exposing the material substance of nature, his paintings also reveal its mystical aspect, its ability to transform in accordance with the cyclical, cosmic rhythm of growth, maturation, death, and rebirth.

**Keywords:** Polish art; symbolism; nationalism; Ruszczyc; cultural exchange; Stimmung

## 1. Introduction

In the decades around 1900, and in the wake of an intensive century of nation building (Belgium, Greece, Italy, and Serbia were among the recently formed nations), a tendency to embody metaphors for regional or national identity in the landscape became a common pursuit (Lukkarrinen 2005, p. 22). Artistically rooted in the St. Petersburg Academy milieu and cosmopolitan Russian modernist circles, Ferdynand Ruszczyc (1870–1936) nevertheless drew inspiration from Scandinavians painters, sharing their passion for the emotionally charged beauty of their native landscapes, the genius loci, and also for the mystical and mythic element in nature. He combined the microperspective of familial environs with a universalist vision of the world's eternal metamorphosis.

Northern landscapes, scenes of the Vilnius region and the Nordic nations, the Baltic Sea, and the island of Bornholm take center stage in the painterly geography of Ferdynand Ruszczyc. His are rugged landscapes, mysterious and melancholic, "exuding moody pensiveness and mysticism" (Krakowski 1966, p. 72). No less abundant or vast was the map of his artistic relationships, broadened through visits to Berlin, Brussels, Crimea, Paris, Rügen, Vienna, and southern Sweden.

## 2. Ruszczyc's Formation

The son of a Polish father, Edward Ruszczyc, and a Danish mother, Alvine Munch (no relation to the painter, Edvard), Ferdynand Ruszczyc was born in Bohdanów, a town situated in the former Ashmyany district near Vilnius (in modern-day Belarus). He completed his education at the Imperial Academy of Arts in St. Petersburg (1892–1897), studying under Ivan Shishkin (1832–1898) and Arkhip Kuindzhi (1878–1910). Ruszczyc registered contemporary impulses flowing in from European art centers; his oeuvre constituted a point of transition between centuries and among cultures, artistic traditions, and modernist currents. His paintings, rooted in the direct observation of nature, exhibit rich textures, saturated colors, and impressive spatial effects. They represent monumental visions of harsh northern landscapes as well as fairytale fantasies with a healthy dose of symbolic allusion that encouraged viewers to discover mythic and poetic elements in the native landscape. They evidence the Romantic pedigree of Symbolism in the Nordic countries, where nature became a vehicle for dreams and moods, and also for national identity. In addition, the artist's oeuvre resonates with Expressionist qualities that at the time Ruszczyc painted had not yet been defined as a part of a coherent artistic program but already constituted a palpable force in contemporary artistic circles.

Since 1795, the Polish-Lithuanian Commonwealth had been divided among Austria (with a main city in Krakow), Prussia (with a capital in Posnan), and Russia (with a capital in Warsaw). In the early 1890s, Poles from all over the Russian partition—including Kazimierz Stabrowski (1869–1929) from Lithuania, Konrad Krzyżanowski (1872–1922) from Kiev, Eligiusz Niewiadomski (1869–1923) from Warsaw—studied in St. Petersburg[1] played fundamental roles in Ruszczyc's growth as an artist and in the development of his concept of landscape (see Skalska-Miecik 1984, pp. 45–55). These St. Petersburg Poles constituted a short-lived yet influential artists' colony (Skalsa Miecik 1989, p. 20). Ruszczyc's education there coincided with a period of significant change at the St. Petersburg Academy, when it shifted away from the critical Realism promoted by *Peredvizhniki* artists (The Wanderers) toward a mystical atmospheric landscape lyricism. Ruszczyc—along with the Russian painters Isaac Levitan (1860–1900), Valentin Serov (1865–1911), Mikhail Nesterov (1862–1942), and Mikhail Vrubel (1856–1910)—participated in this trend.

Under Shishkin's watchful eye, Ruszczyc honed his skills as a draughtsman. To his Russian mentor, an artist with roots in the Romantic landscape tradition of the Dusseldorf Academy who painted primarily panoramas, wooded rural scenes, and wilderness, Ruszczyc owed his mastery of chiaroscuro and his adroitness in crafting mood. He also admired the work of Levitan (Ruszczyców na 1966, p. 15), who originated a masterful synthesis of landscape through cropped framing and close-up views of fields, meadows, blossoming orchards, forest trees, deserted trails, wetlands, and skies with contrastingly illuminated clouds. The intimacy of these nature visions emerged in the sketch-like, broad rendering of motifs and in visible brush strokes of thick paint that mingled with occasional smooth spots. In some of Levitan's panoramas a brisk sky mottled with heavy clouds weighs on an open landscape vista. Tillage is another frequent motif, with human silhouettes dwarfed by the apparent limitlessness of freshly ploughed fields. Yet, more than anyone, Arkhip Kuindzhi (1842–1910), whom Ruszczyc revered (and who reciprocated those feelings to his talented student[2]), ingrained in him a deeply emotional and subjective understanding of nature. Distinguished by a bold, broad painting style that pushed the boundaries of conventional compositional structure and shunned detail and description, Kuindzhi's mysterious landscape visions, evidence a spirit similar to that of the Symbolist poets of the time and made a particularly strong impression on Ruszczyc.

Ruszczyc magnified the extraordinary concentration of expression in his works through high horizons, close-ups, and fragmented angles that obstructed a comfortable sense of space. These evidenced a modernist urge to negate an image's depth and mimetic possibilities in order to tap into the subjectivity of a 'landscape of the soul'. The emotions and thoughts engendered by the world of phenomena took precedence over precise description. To enhance the expressiveness of his paintings, Ruszczyc utilized intense, saturated colors

and thickly applied paint that register his energetic brush strokes, an effect evident in the vigorously, fluidly, and thickly painted *Cloud* (1902, National Museum in Poznań), where objects are reduced to nearly abstract signs and fields of vibrant color.

### 3. Between Intensivism and Symbolism

In 1897, the contemporary art critic Cezary Jellenta coined the term "Intensivism" in an article published in the journal *Głos* to describe Ruszczyc's art (Jellenta 1897, pp. 819–25, 835–36, 843–47, 870–73). Intensivism described art that moved the viewer emotionally and reflected the spiritual state of the artist, his creative desires, impulses, and visions. Intensivism, which strove to attain the expressiveness of nature's vigor by eliminating redundant formal elements, finding a singular synthesis, and emphasizing main motifs, reconciled the aims of Symbolism with the budding movement of Expressionism.[3] According to Jellenta, the main premises of Intensivist art involved eschewing illustration in favor of innuendo and applying a specific approach to framing that brought viewers closer to nature and guaranteed strong sensations: "the painter does brutal work on us, seizes us and yanks us one way, shakes us with one strong impression, hypothesizes an irresistible suggestion" (Jellenta 1897, p. 822) and imbues "the viewer's soul with tragedy, with a sublime gloom and dark brilliance that cannot be described" (Ibid., p. 824). For Jellenta, Arnold Böcklin (1827–1901) and Anders Zorn (1860–1920) produced archetypal landscapes in this mode packed with dramatic power.

Ruszczyc admired the work of Symbolist pioneer Böcklin, whom contemporaries dubbed "a strange meteor in modern art" (Witkiewicz 1884, p. 500). He surrendered himself to the impulses he detected in Böcklin's paintings (seen at exhibitions in St. Petersburg), especially in the paintings and drawings that invoke the world of Mediterranean mythology in works like *Faun* (*Evening Star*) and *Tritons* (*In the Depths of the Sea*) from 1897 (lost). The Swiss artist's formula of combining a mysterious and foreboding landscape with bold colors utilized as a vessel for communicating mood, were shaped by modernist impulses then beginning to surface. Böcklin painted pictures of perilous waters that leave the viewer shaken, as well as sacred groves inhabited by poetic creatures that exude the elementary forces of nature, "a visual symbolization, modelled on Ancient Greek art, of the moods of nature and the artist" (Feldman 1897). Böcklin's pantheism resonated strongly with artists searching for ways to express either their spiritual conditions or the soul of the world.

In Polish art criticism of the late nineteenth century, the fashionable term for moodiness— the German term *Stimmung*—became a category that bridged the traditions of Romanticism and Symbolism. As Polish painter, philosopher, and writer Stanisław Witkiewicz declared: "And so, mood [. . .] represents the freshest direction in art and the best attribute in all things" (Witkiewicz 1901). Chiaroscuro effects, subtle tonal gradations, and unexpected color harmonies built particular landscape "modi," "from dreadful drama to tranquil idyll" (Witkiewicz 1884, p. 497), and were intended to make a strong impact on the viewer. During Ruszczyc's 1898 travels through Austria, Belgium, France, Germany, Italy, and Switzerland, he experienced a "moment of great joy" when viewing Böcklin's work (Ruszczycówna 1966, p. 34). The Symbolist's paintings captivated him with their effects of somnolent stillness, the mysteriousness and silence of moonlit scenes, and the artist's adroitness in combining observation of nature with pure imagination.

The luminous canvases of French Impressionist Claude Monet (1840–1926) that Ruszvzyc saw in Paris were another revelation. In an 1898 journal entry, the artist wrote: "Looking at those melting pigments, at that air so curiously rendered, I felt many things that I had never known before" (Ibid., p. 33). Russian critics, however, tended to be equally fond of Nordic painters for the poetic suggestiveness of their visions.

Beginning in the mid-1890s, Russian artists' contact with artistic centers in Western Europe and Scandinavia intensified. This is evidenced by striking analogies, for instance, between paintings by Ruszczyc and those of the Finnish painter Akseli Gallen-Kallela (1865–1931) or the Swede Anders Zorn, both of whom were popular in Russia. The Polish artist also admired the Norwegian Impressionist Frits Thaulow (1847–1906), as Ruszczyc

noted in an 1897 journal entry (Ibid., p. 27). Ruszczyc had opportunities to encounter the paintings of Danish, Finnish, Norwegian, and Swedish artists in St. Petersburg as well as in Vienna, at an exhibition staged by the Vienna Secession in late 1901 that continued through early 1902. In particular, Western European and Russian modernist circles received Gallen-Kallela, dubbed a "cosmopolitan nationalist" and active during a period of intensified russification in the Grand Duchy of Finland, with equal enthusiasm (Walkowiak 2013, p. 461). In Russian modernist circles, interest in the Nordic nations—their literature, theater, and fine art—was "so widespread and deep-seated that it constituted almost something of a cult" (Konstantynów 1996, p. 162). From the 1880s on, Danish, Finnish, Norwegian, and Swedish art appeared regularly at exhibitions held in Berlin, Munich, and Paris. A pro-Nordic attitude characterized the magazine *Mir Iskusstva* (1898–1904), founded by St. Petersburg painters and intellectuals centered around the impresario Sergei Diaghilev (1872–1929) and the painter Alexandre Benois (1870–1960). As the periodical's name (The World of Art) suggested, its founders believed in the autonomy of art as well as in the confederation of its various disciplines in the spirit of Richard Wagner's fashionable idea regarding the original unity of painting, music, and literature.

This revolt against positivist utilitarianism and the cult of individuality led to a break from naturalistic genericism and an embrace of poetic and subjective realities. Unlike the previous generation that cultivated Russian "intellectual-artistic isolationism" (Cieślik 1986, p. 43), Symbolists expressed an enthusiastic embrace of Western poetry and philosophy. By publishing writings authored by an international coterie—German composer Richard Wagner (1813–1883), Polish writer Stanisław Przybyszewski (1868–1927), Belgian poet Maurice Mæterlinck (1862–1949), and English writer John Ruskin—it became the chief organ of Russian modernists and a forum for the presentation of Western European art and ideas. As early as 1897, Diaghilev organized an exhibition of work by Danish, Norwegian, and Swedish artists (including the Swedes Carl Larsson, Bruno Liljefors, and Zorn, and the Norwegians Thaulow and Edvard Munch) at the Imperial Society for the Encouragement of Fine Arts in St. Petersburg (Konstantynów 1996, p. 163). The following year, *Mir Iskusstva* brought an exhibition of work by Finnish and Russian artists to St. Petersburg (Sinisalo 2005, p. 19), though joint shows had been held earlier. Enjoying particularly strong critical admiration were the Finns Albert Edelfelt (1854–1905) and Gallen-Kallela. In 1899, the magazine's editors staged the International Exhibition of Paintings, which, alongside works by Böcklin and Russian and French artists, featured paintings by Finns and Norwegians (Konstantynów 1996, p. 165). Gallen-Kallela's landscapes challenged the rules of Naturalism and seemed to capture the spirit of the mystical North, especially his desolate panoramic winter scenes and cloudy sky compositions, characterized by synthesis, simple chiaroscuro modelling, interplay of cool planes, pearly-grey colors, and a decorative treatment of natural motifs such as trees and reflections in water. "It would be wise to introduce Warsaw to Finnish art, or Scandinavian art in general," Ruszczyc wrote in 1903 (Ruszczycówna 1966, p. 48).

In his journal, Ruszczyc expressed a need to paint things he sensed, things from the domain of his inner experience: "To this land of mine, for I have nothing more valuable [...] I offer my feelings" he wrote, citing Polish art critic Czesław Jankowski (1857–1929) (Ibid., p. 47). His paintings often depict intimate childhood scenes from his family homestead near Vilnius. Despite their dream-like qualities, works like *Mill* (Figure 1) and *Last Snow* (1898–99, National Museum in Warsaw) also function as patriotic symbols of his Polish homeland, as does *Forest Stream* (Figure 2), which shows an area in the park in Bohdanów described as "huge, full of towering trees, thickets of wild brush, sunny clearings and hidden trails, and ended at a winding brook in an alder grove steeped in an otherworldly solitude" (Bułhak 1939, p. 92).

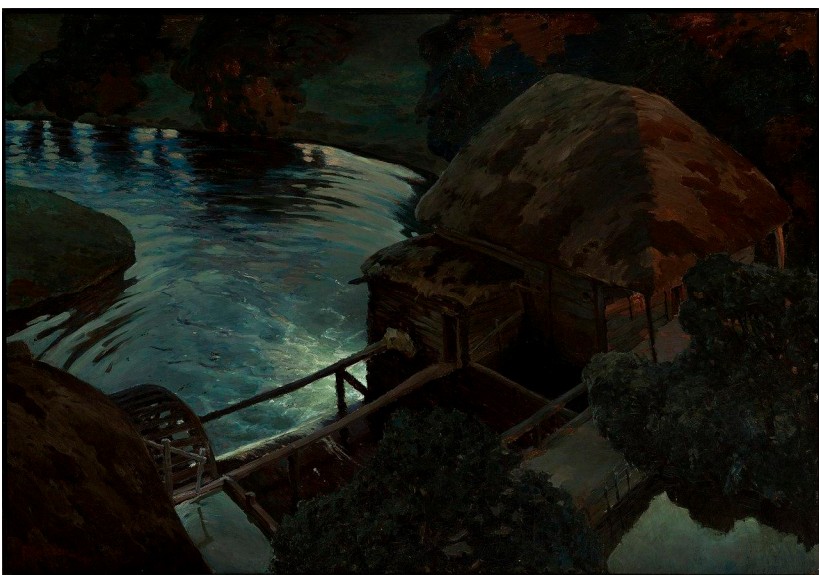

**Figure 1.** Ferdynand Ruszczyc *Mill*, 1898. Oil on canvas, 112 cm × 162 cm. National Museum in Krakow.

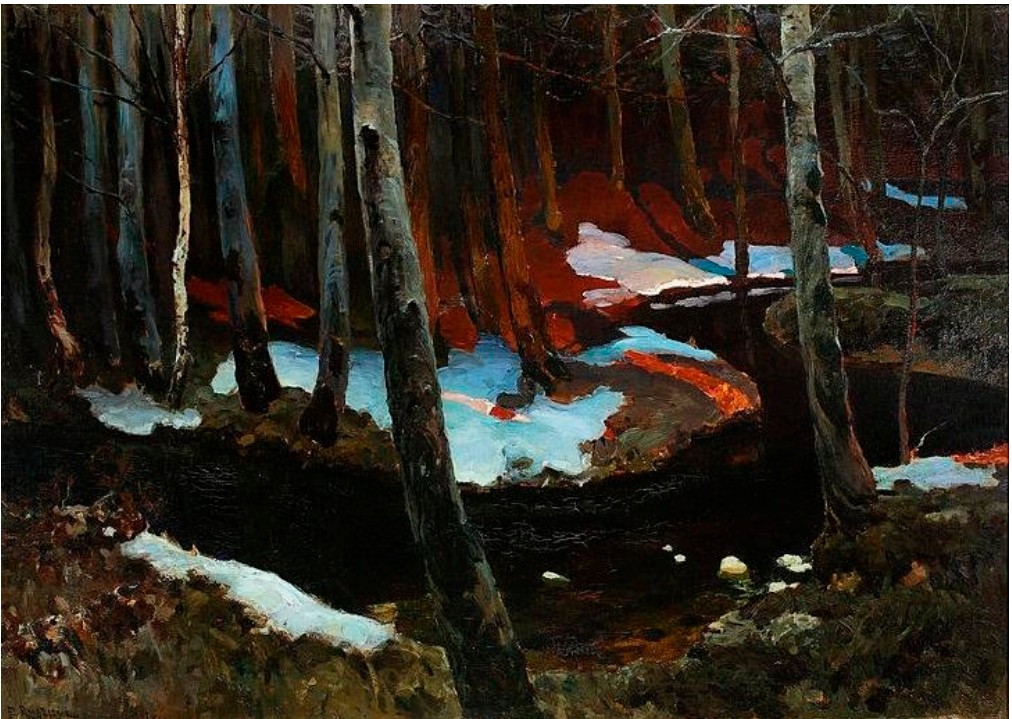

**Figure 2.** Ferdynand Ruszczyc, *Forest Stream*, 1900. Oil on canvas, 113 cm × 160 cm. National Museum in Warsaw.

As evidenced here, changing seasons and natural states, elements, elementary substances, cloud and soil formations, the depths of the sea, and solstices fascinated Ruszczyc. Following the example of Japanese artists, he captured the beauty of minute natural elements seen up close: tree trunks, dainty, decoratively stylized branches, or streams (as in *Autumn Landscape with Setting Sun*, 1907, National Museum in Warsaw). The 'tight' framing conveys the artist's random 'snapshot' gaze and underscores his identity with the observed scenery. Ruszczyc's landscapes often portray raw earth and bare tree trunks casting long shadows. These evoke a mysterious emptiness, a moment of dying away. They showcase

both the material substance of nature and its mystical side, its capacity to transform in accordance with the cyclical, cosmic rhythm of growth, maturation, death, and rebirth.

*Earth*

*Earth* (Figure 3), painted in the autumn of 1898, occupies a special place in the artist's ouevre. A large-format canvas split into two zones—a cloudy, dynamically painted sky and a hilltop of dark, barren, furrowed soil that obscures the horizon.—it evokes a hump, a strange growth, and functions as an ominous obstacle to the viewer's gaze. Ruszczyc, utilized a dramatic contrast between light and dark to emphasize the discrepancy between these zones. As art historian Wojciech Suchocki notes, the painting's surface and its boundaries actively contribute to the construction of meaning (Suchocki 2004, p. 20). Amid clashing elements, the artist placed a small figure of a hunched ploughman driving a pair of oxen. This monumental painting depicting the ties of humans to nature, human toil, and the farmer's labor, as well as to life's cyclical rhythm, radiates a massive expressive force and makes an arresting painterly impact. Thanks to photographic documentation by Jan Bułhak, we know how this landscape looked in real life (Bułhak 1939, p. 92). It was a gentle knoll blanketed with forest on its left, with a row of firs on the right. By eliminating the trees and monumentalizing the hilltop, the artist achieved a synthesis and compounded the impression of the earth's majesty as he stepped beyond the boundaries of Naturalism.

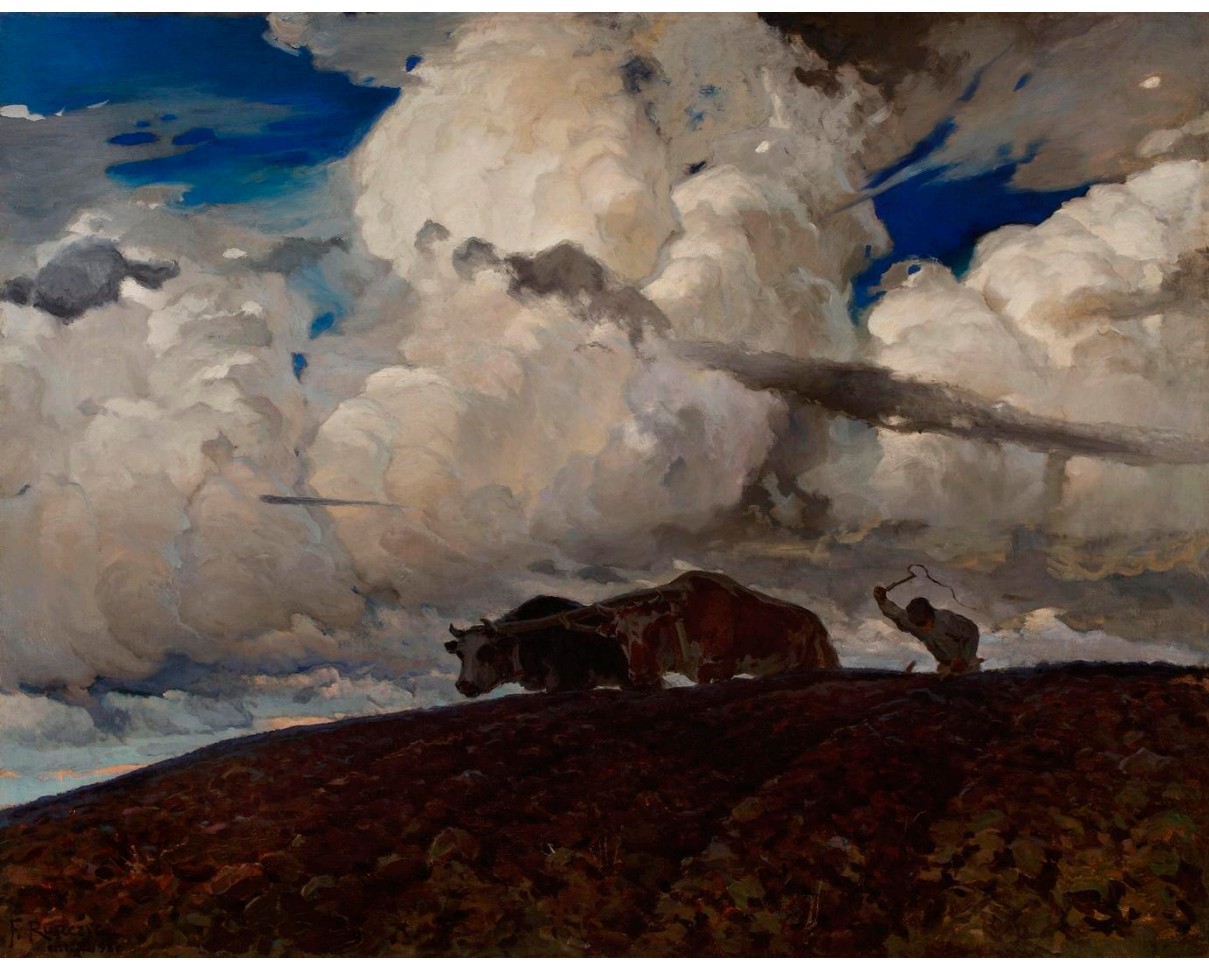

**Figure 3.** Ferdynand Ruszczyc, *Earth*, 1898. Oil on canvas, 164 cm × 219 cm. National Museum in Warsaw, inv. no. MP 393 MNW.

During an exhibition in Vienna, critics compared Ruszczyc's paintings with the work of artists of the Worpswede colony, especially Otto Modersohn (1865–1943) (Hevesi 1902). *Earth*, enjoy great critical acclaim in its day and established Ruszczyc as a major Polish landscapist. The painting became a symbol of the nation's destiny (Morawińska 1997, p. 108), an allegory of Poland, and an artistic and ideological declaration (Ibid., p. 108), particularly since the artist had used the same motif for his cover design for the Parisian journal *La Pologne Contemporaine*.[4] The depiction of a heroic struggle with nature, however, also situated the painting in the sphere of biblical and mythological symbolism relating to the Fall of Man. It points to God's judgment as described in the Book of Genesis—the eternal punishment for humanity's Original Sin—and also to imagery of the 'Iron Age', a stage in world evolution described by Hesiod in *Works and Days* and by Ovid in *Metamorphoses*, when the gods left, and humanity could only dream of its lost *aurea ætas* (eternal spring), a time of plentitude and joy, when the generous earth, untouched by the plough, benevolently yielded its bounty.

When addressed by Naturalist artists, the subjects of arduous labor, exploitation, and social degradation usually invited reflection on gloomy contemporary conditions, the anticipated collapse of human society, and the disintegration of culture so aptly diagnosed by Friedrich Nietzsche. From such a perspective, monotonous and lifeless fields could therefore be interpreted as an 'anti-Arcadia': landscapes intentionally anti-ideal or anti-pastoral. Yet, the active figure of the ploughman placed at the crest of the hilltop and silhouetted against the illuminated firmament compels us to see the painting as a visual manifestation of Nietzschean triumph of the will. The philosophy of the German thinker, who proclaimed the cult of strength and life, the idea of superhumanity, a disdain for weakness, and the necessity to conquer death, became a point of reference for Russian modernists in their deliberations on philosophy and art. Here, the ritualized work of ploughing constitutes an act of creation, a demiurgic gesture of sowing, a revelation of creative energy. Mieczysław Limanowski argues that this painting can be considered a work of religious art that references the Eleusinian myth, the story of Demeter and Persephone, in addition to incorporating Christian symbolism of death and resurrection (Limanowski 1939, p. 399).

Another of Ruszczyc's paintings, *Sobotki* (Figure 4), "symbolising the vitality of people cultivating Proto-Slavic customs" (Kossowska n.d.), also refers to worship of chthonic deities. The substance of the earth, rendered with coarse impasto, seems tangible and heavy, as do the dense fluffy clouds, which fill the space of the painting in a manner that conveys an impression of motion, of the sky pressing down on the land. Ruszczyc conveyed the materiality of the earth, its primordial substance, with pulsing chromatics, that reflect its geological durability and longevity. Focus on the structure of the soil raises questions regarding the constitution and age of the planet, the processes transpiring on it, humanity's changing relationship to the earth, and the interdependence of the human species with nature in an era when science challenged Biblical perspectives and questioned old systems of belief that asserted humankind's central position in the universe (Thomson 2012, pp. 127–28). As Michał Haake points out, Ruszczyc was influenced by the paintings of Claude Monet he had seen in Paris at Galerie Georges Petit in 1898—images of masses of earth and studies of the cliffs in Val-Saint-Nicolas near Dieppe and Petit Ailly—that betrayed a fascination with geology and Darwin's theory of evolution (Haake 2018). Yet, scientific undercurrents fail to explain *Sobotki*'s utterly strange depth, in which a sky with luminous clouds towers over ridges of black earth, as if a vision of infinity were opening above the horizon of human cognition. The painter Eligiusz Niewiadomski (1869–1923) wrote that "out of an ordinary ploughing scene, Ruszczyc created a wonderful Symbolist drama. Beneath the immense sky billowing with a mass of formidable clouds, Mother Earth the Provider quietly arches her back" (Niewiadomski 1926, p. 312). Mother Earth, sacred Gaia, thus also becomes a universal symbol of life, with an innate antinomy of matter and spirit. Artists of the Young Poland movement (*Młoda Polska*) often made paintings that omitted the sun, thereby emanating a sense of emptiness. Ruszczyc reached the pinnacle

of expressive Symbolism by imbuing emptiness with a sublimity and light evoking the mysticism of the North.

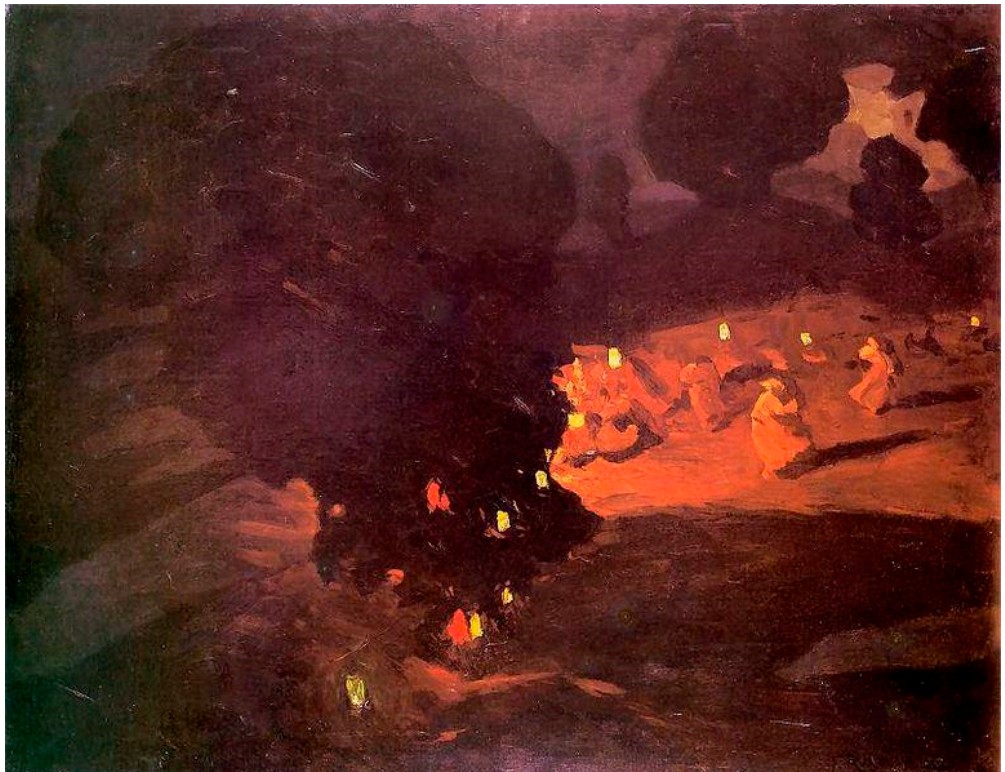

**Figure 4.** Ferdynand Ruszczyc, *Sobotki*, 1898. Oil on canvas, 72 cm × 92.5 cm. Częstochowa Museum.

*Old Apple Trees* (Figure 5) and *Old House* (Figure 6) also exhibit profound emotion and patriotic symbolism, qualities indicated by their titles, which communicate the value ascribed to these venerable places and to the endurance of the artist's memory and imagination. Toward the end of 1901, Ruszczyc even entertained the idea of making a triptych (never realized) illustrating the life stages of a single tree—an apple tree, as a *pars pro toto* example of nature at large and a metaphor for its eternal revival of strength in order to communicate the resiliance of Poles and their determination that the partitioned Polish nation will one day be reunited, a dream realized in 1918. The orchard in Bohdanów, with its mysterious trail that meanders into the wilderness, becomes a curious temple, a Baudelairian "forest of symbols," an abyss of dreams and nightmares. The bare, convulsively twisting branches create organic structures and abstract rhythms evocative of the whiplash ornamentation of Jugendstil design. The trunks, seemingly stripped bare by illness yet still decorative, contrast with the vibrant green of the grass, where a black pit lurks in the plump soil. Ruszczyc portrayed nature as if in a moment of painful tension, in anticipation of spring exploding, of its resurrection, like that of the Polish nation.

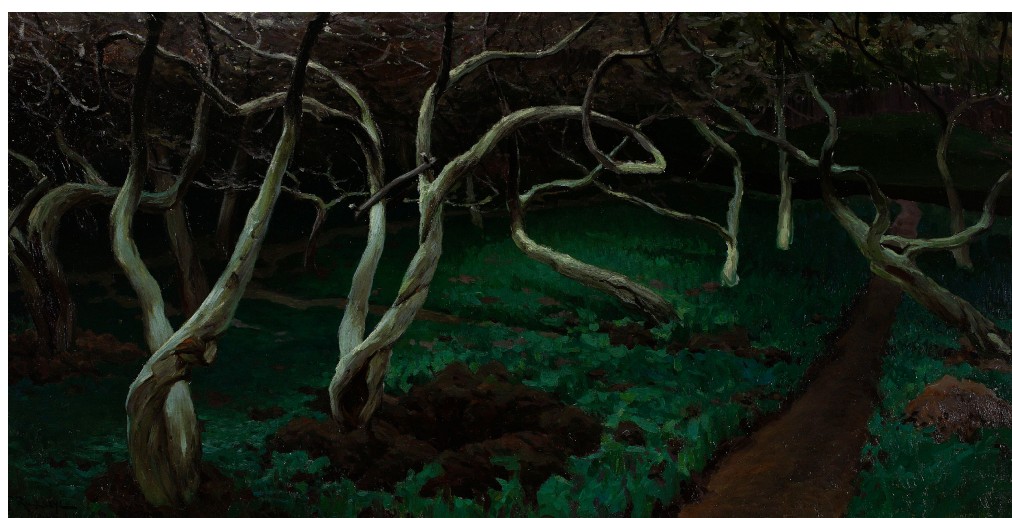

**Figure 5.** Ferdynand Ruszczyc, *Old Apple Trees*, 1900. Oil on canvas, 85 cm × 165 cm. National Museum in Warsaw, inv. no. MP 394.

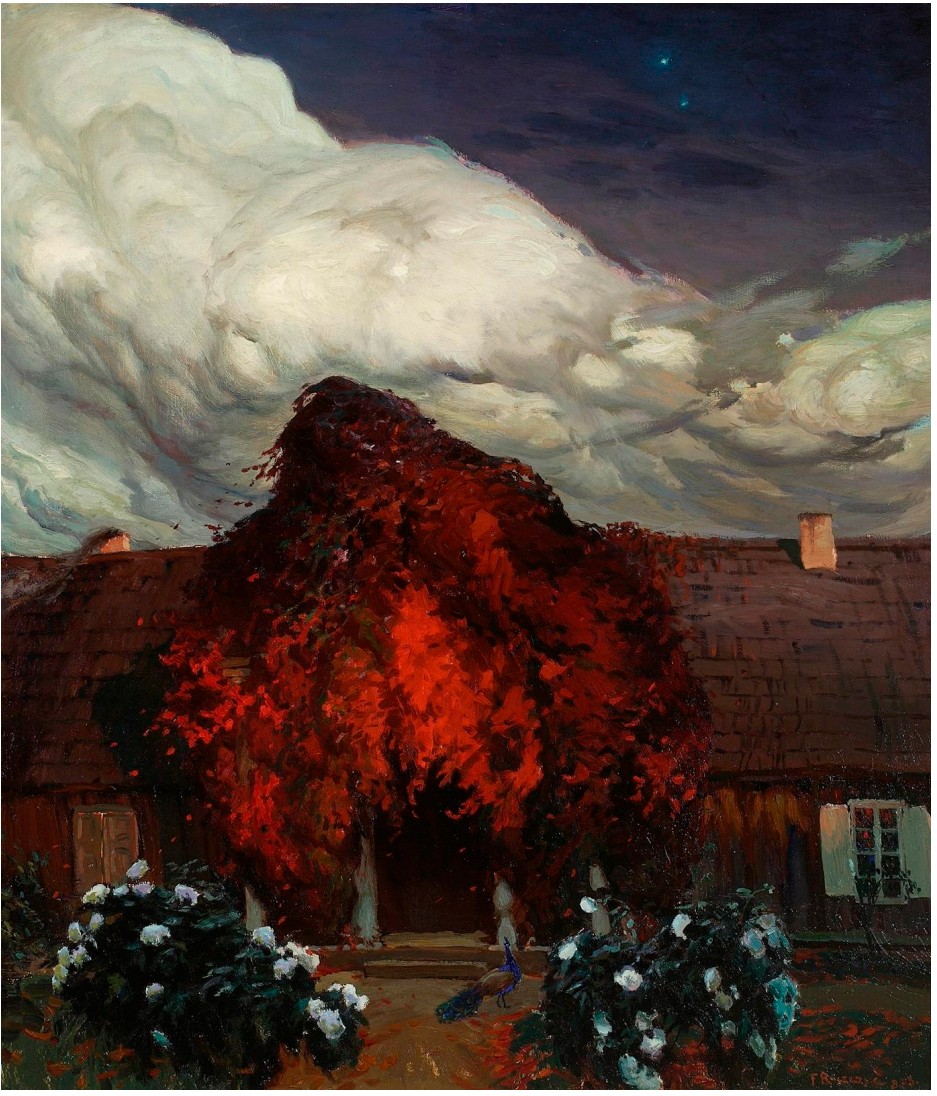

**Figure 6.** Ferdynand Ruszczyc, *Old House*, 1903. Oil on canvas. 93.5 cm × 83 cm. National Museum in Warsaw, inv. no. MP 392 MNW.

## 4. In Search of National Identity and Dialogue with European Art

*Old House*—the wooden seventeenth-century manor in Bohdanów where Ruszczyc grew up and whose neglected porch become overgrown with lush grapevines—is intimately tied to his personal identity. Here, he revealed his attachment to his native land and rootedness in the geography of his ancestors. Although for Polish viewers this clearly represents an upper-class family home, in the nineteenth century, when the Polish nation was occupied, such noblemen's manors represented bastions of Polishness, symbol of enduring tradition, and psycho-emotional security. In an era of technological advancements, rapid modernization, overcrowded metropolises, and uncertainty regarding the universe's longevity, the family home became an oasis, a family Arcadia, a treasury of national mementoes, and a place of patriotic education, especially in the context of the country's loss of statehood in the eighteenth century and the threats and repression of tsarist authorities after the January Uprising (1863) in the lands of Lithuania and Congress Poland. In Ruszczyc's interpretation, the home became a living organic structure providing shelter from the elements and the darkness of night, actual and metaphorical.

Though Ruszczyc settled in Bohdanów after graduating from the St. Petersburg Academy, he maintained contact with it and participated in three consecutive Spring Exhibitions there. From 1899 onward, he participated regularly in *Mir Iskusstva* exhibitions.[5] Artists in that circle explored with great eagerness themes like the legendary history of the Rus' region and mythic and folkloric motifs, often subjected to a formula of decorative Synthetism or Jugendstil and folk styles. Several spectacular paintings by Ruszczyc reflected such imaginative and poetic directions, including *Nec Mergitur* (Figure 7) and *Winter Tale* (Figure 8).

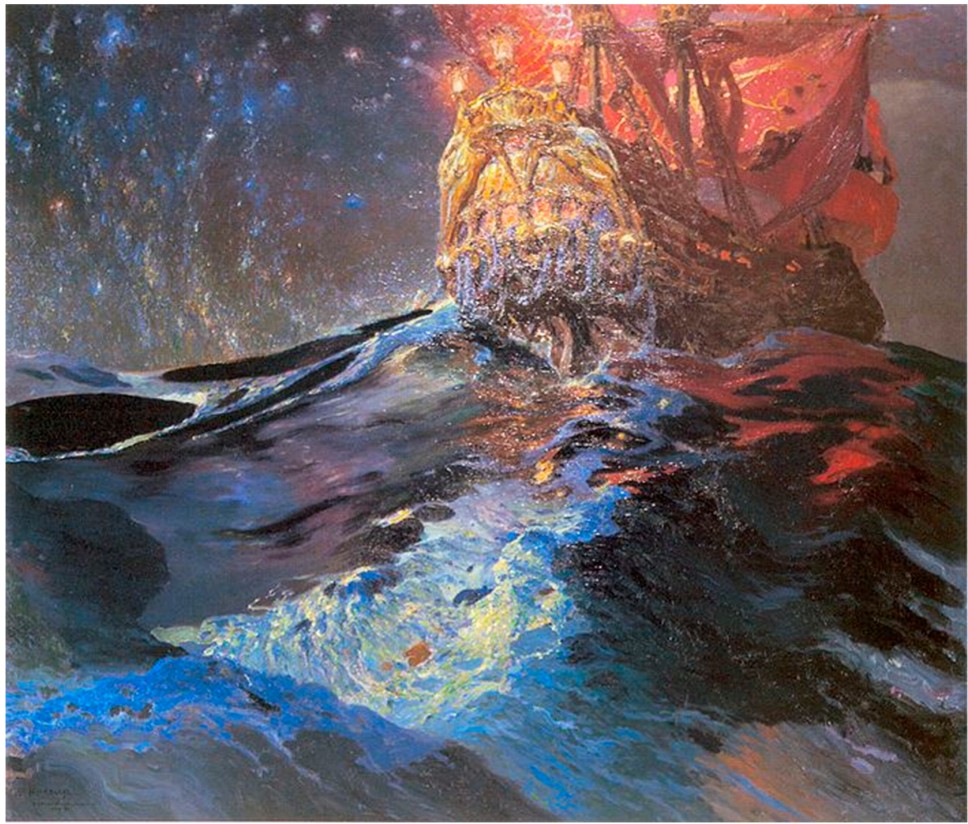

**Figure 7.** Ferdynand Ruszczyc, *Nec Mergitur*, 1904–05. Oil on canvas, 204 cm × 221 cm. Lithuanian Art Museum, Vilnius, inv. No. T 2691.

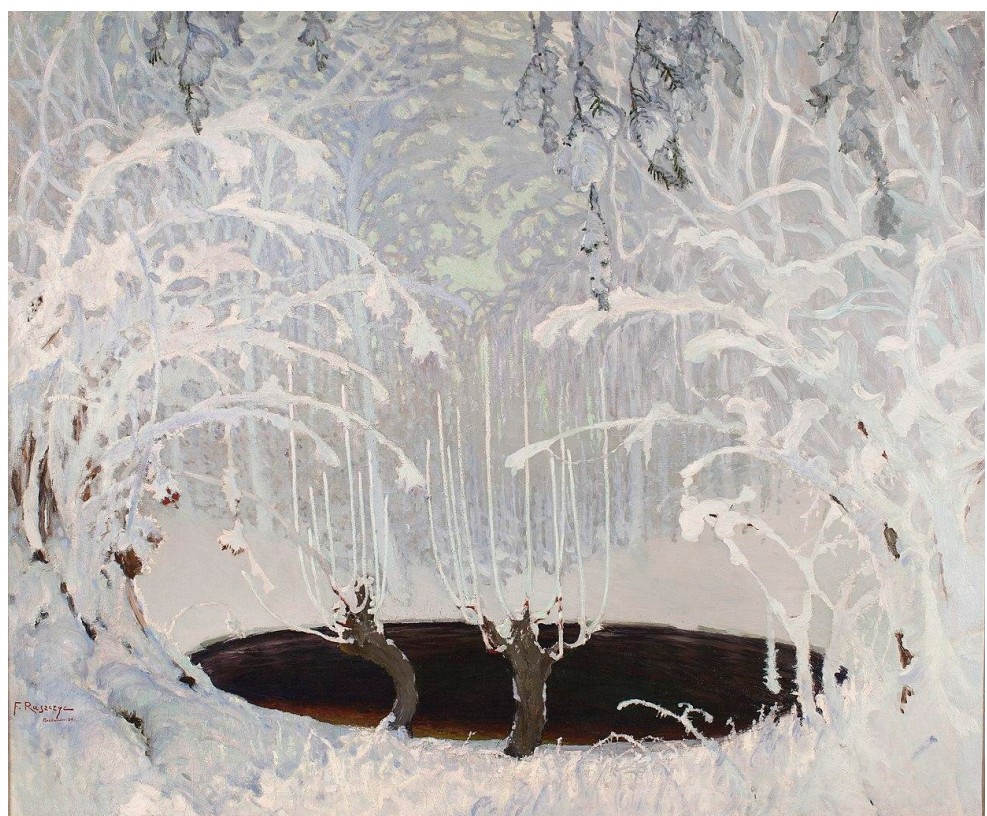

**Figure 8.** Ferdynand Ruszczyc, *Winter Tale*, 1904. Oil on canvas, 132 cm × 159 cm. National Museum in Krakow.

*Nec Mergitur* (see Morawińska 1988, pp. 145–57), inspired by the motto on the coat of arms of Paris (*Fluctuat nec mergitur*) and considered Ruszczyc's most visionary work, abounds in patriotic symbolism because it portrays a Polish royal naval vessel rocked by German and Russian waves (Niewiadomski 1926, p. 312). Inspiration for the painting is said to come from the artist's reading of Henryk Sienkiewicz's "A Legend of the Sea" (1900), a story about the dramatic fate of a ship named *The Purple* that Sienkiewicz treated as an allegorical tale of the reasons for the fall of the Polish-Lithuanian Commonwealth. Yet, this imaginary masterpiece veers far from a literal illustration of the patriotic story. Against a cascade of stars pouring down from the sky, the ghost ship in the painting, which, as the artist himself wrote, was "unreal" (Ruszczycówna 1966, p. 49), becomes a surrogate for human fate: it points to the literary topos of *navigatio vitae*, the voyage of the Argonauts, the journey of Odysseus, Arthur Rimbaud's *The Drunken Boat*, as well as to Samuel Taylor Coleridge's ballad *The Rime of the Ancient Mariner*, illustrated by Gustave Doré (Morawińska 1988, pp. 145–57). In the visual dimension, the painting engaged in a dialogue with Romantic marine painters like J.M.W. Turner (1755–1851) and Eugène Delacroix (1798–1863). The impact of Ruszczyc's vision lies in the force of the tempestuous elements he portrayed and the majesty of the firmament, in the sense of limitlessness and nature's cosmic power. The lucent rocking galleon seems to drift in an abyss of infinity or in a fairytale realm. Interestingly, the motif of a red-sailed ship on dark blue waters depicted in an extremely shortened perspective appeared in *Guests from Overseas* (1901, Tretyakov Gallery, Moscow) by Nicholas Roerich (1874–1947), another student of Kuindzhi.

The poetically titled *Winter Tale*, shows an unrealistic forest blanketed in white hoar frost that resembles an arabesque verging on abstraction, thanks to the rhythm of the decorative, Jugendstil branches. Here, the magic of the northern landscape emerges in the symphony of whites, in the fantastical mirage of calligraphic lines and trickles of paint, and in the enigmatic and impenetrable depth of the black reflectionless pond. The lifeless

and bare surface of the water—a universal symbol of passage and death but also of the human soul—resembles a mysterious dark portal into transcendence. *Winter Tale* depicts a world without humans in which nature persists in its untouched eternal beauty, subject only to the aesthetic laws of painterly representation. Finnish painter Victor Westerholm (1860–1919) adopted a similar aesthetic approach in winter scenes such as *Landscape of the Kymi River* (Figure 9).

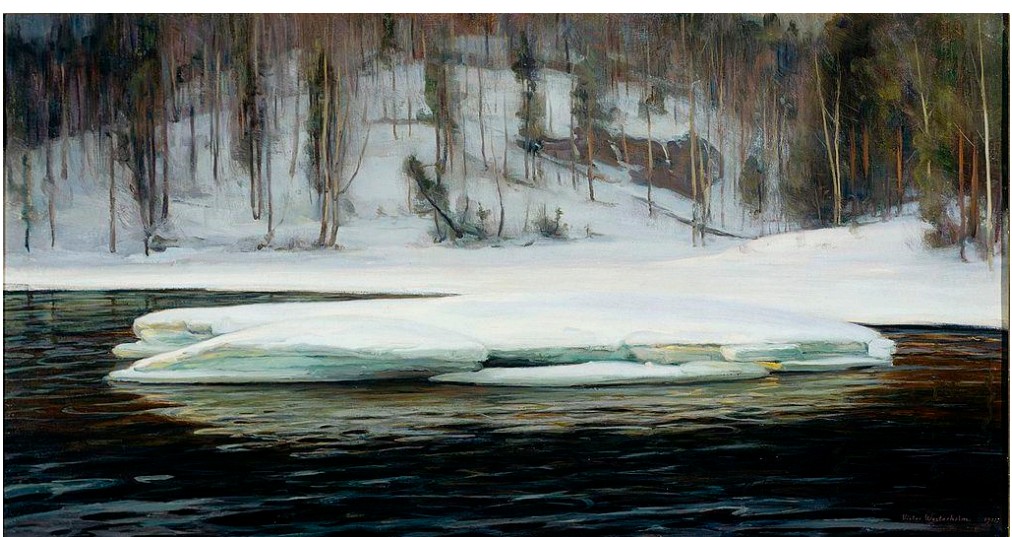

**Figure 9.** Victor Westerholm, *Landscape of the Kymi River*, 1901. Oil on canvas, 58 cm × 108 cm. Turku Art Museum.

## 5. Conclusions

The career and oeuvre of Ferdynand Ruszczyc exemplified how painters in the decades around 1900 met the challenge of creating works that in their subject and style functioned simultaneously as deeply personal and philosophical, as exemplars of patriotic sentiments, and as evidence of an earnest engagement with progressive, contemporary, international aesthetic tendencies. Like his colleagues stationed elsewhere around Baltic shores, Ruszczyc strove to create profound works that expressed his individual identity at the same time as they functioned in a larger, national and international, realm by effectively communicating to his compatriots the Polish trauma of partition and hope for reunification, and to fellow world citizens, anxiety about the future and nostalgia for the past felt by many during that dynamic and unstable era.

**Funding:** This research received no external funding.

**Conflicts of Interest:** The author declares no conflict of interest.

## Notes

1    From late 1795, the time of the third partition of the Polish-Lithuanian Commonwealth between Russia, Prussia and Austria, a vast expanse of Polish land, including the capital city of Warsaw, became part of the Russian Empire.

2    Under the guidance of his professor, Ruszczyc worked on competition paintings at the academy. The 1897 show was something of a manifestation of the formation known as the Kuindzhists and it brought Ruszczyc much praise and interest from collectors: (Pavel Tretyakov bought the painting *Spring*. (Mytarewa 1966, pp. 59–60).

3    Tadeusz Lewandowski describes his work as early, *avant la lettre* Expressionism. (Lewandowski 1971).

4    (Krakowski 1966, p. 73). C.f.: https://cyfrowe.mnw.art.pl/pl/katalog/781177 (accessed on 7 March 2021).

5    In 1900, the artist was invited to become a member of the *Sztuka* Association of Polish Artists, and in 1907, he was appointed the Association's chairman. He also showed his work at the Zachęta Society for the Encouragement of Fine Art in Warsaw, with much success. After several years in Warsaw (1904–07), where he taught at the School of Fine Art, he took over as the head of the landscape painting department (1907/08) of the Academy of Fine Art in Krakow. In 1908, he moved to Vilnius. There, he gave up oil painting and devoted himself to social and education work; he designed posters and did applied graphic work while also

advocating for the preservation of Vilnius Region cultural heritage. As a stage set designer, he staged a number of Romantic works by the great national writers, and in doing so, referenced Slavic history. He also served as the first dean of the Faculty of Fine Art, established in 1918–19, at Stefan Batory University in Vilnius.

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
