# Peer review of "Ferdynand Ruszczyc: A Polish Painter at the Crossroads of Cultures"

_arts, 1900_

Round 1

Reviewer 1 Report

Comments and Suggestions for Authors

The proposed article is interesting and its topic falls perfectly within the scope of the journal. It shows a good knowledge of the work of the artist in question. The analysis of the paintings and their contextualization are good and measured. It seems to me, however, that they can be improved and expanded, which will only strengthen the author's thesis.

I think the major problems here are related to the structuring of the text rather than any flaws in the research. I would advise the author to revise the abstract and the introductory part of the article. Now they are dominated by extensive historical descriptions and it is difficult for the reader to understand what the article actually concentrates on. For example, in the abstract, the purpose of the article is explained only in the last sentence. The introduction is structured in the same way, and the article should begin with a clear and concise description of the aims, objectives and thesis of the study.

The conclusion at the end is also too short. I believe that in the interest of clarity of research, it should be expanded.

Comments on the Quality of English Language

The text is written in very good English, but the influence of the author's native language is evident, sometimes in the word order and in most cases in the vocabulary used. It would be good if the author has the opportunity to find help from a philologist who will enrich the language of the article, polish it and make it more refined. 
It is not a big problem though.

Author Response

As per the reviewer's suggestions, both the abstract and introduction have been modified to make clear the article's purpose. A conclusion has been added and a professional native English speaking editor has transformed the awkward parts of the text into scholarly American English.

Reviewer 2 Report

Comments and Suggestions for Authors

The article examines the work of Ferdynand Ruzczyc as an artist active at the crossroads of different cultures and artistic tendencies. The author argues that his art includes a mystical face of nature and evokes the Romantic landscape and in both respects demonstrates interaction with Nordic visual culture. At the same time, his landscapes include a patriotic content and national identity. The first argument is well supported by visual analysis and by written source, for example when mentioning that Ruszczyc wrote about his reverence for Finnish art and suggesting that it would be “wise to introduce Warsaw to Finnish art, or Scandinavian art in general” (line 162). However the visual evidence, upon which much of the argument is based, only appears on p. 5 of the article, after written description of artistic tendencies. It could be helpful to the reader if some images were added earlier into the paper to help follow and asses the argument. For example when mentioning the works Mill, Last Snow and Forest Stream for the first time on lines 169-170 it could be helpful to add small images of these works.

Another point is presenting more information on the paintings that are integrated into the text would support the argument, for example regarding Earth (Fig. 1) it would be helpful to have dimensions- the size of the image is pertinent to the analysis as is the artistic medium (also not mentioned in the figure details). There is a long paragraph describing another of Ruszczyc’s paintings, Sobotki but with no image of it inserted it is hard for the reader to assess whether the author’s analysis is convincing from a formal point of view, we either need to take his/her word for it or search for the image independently. In my view this harms the flow of argument. This work is found online in the Public Domain so there seems no technical reason not to include it in the article. Regarding the main example, Earth which also gives the article its title, it would be helpful to add the of the same landscape by Jan BuÅ‚hak (mentioned on line 196) to show the unique characteristics of the painting and the way it interpreted rather then just presented the local landscape. 

Apart from the influence of Northern mysticism on Ruszczyc’s art the author also argues for patriotic symbolism found in his works. In my view this point should be explained in more detail to the reader not familiar with the immediate cultural context. For example on line 257-258 the author writes: “Old Apple Trees (Fig. 2) and Old House (Fig. 3) also exhibit profound emotion and patriotic symbolism, qualities indicated by their titles, which communicate the value ascribed to these venerable places and the endurance of the artist’s memory and imagination”. For me the very descriptive titles do not raise these associations, and it may be prudent to explain the interpretation further either in the text or by supporting references explaining the link between the trees and the house with patriotic symbolism. There is more detail about the house motif on the next page but with no references to show that these currents are wider then the author’s subjective interpretation.  

In the conclusion of the article the author re-iterates the two main arguments: 1. That the artist was rooted in the tendency of the turn of the twentieth century for searching for metaphors of regional and national identity in landscapes. 2. That Ruszczyc drew equal inspiration from the St. Petersburg Academy milieu and from the art of the Scandinavians, especially in the mystical and myth element of nature. It is the second point that is better substantiated in the paper, especially through visual analysis. The first point would benefit from being better contextualized with visual or literary evidence. If the author could expand and explain local politics and how the works of art contribute to patriotic ideals expressed in writing or local visual culture it would widen the impact and value of the article and make the analysis more well-rounded between the two complementary arguments. 

Finally two very technical comments. 

  • Some artists mentioned have their life years in brackets, others do not. It would help the reader if artists were presented in a conform way.   
  • There is a typo/ English error on p. 6. In the following sentence ‘enjoy’ should be in past tense I think (enjoyed): “Earth, en joy great critical acclaim in its day and established Ruszczyc as one of the major Polish landscapists” (line 202-203).  

Author Response

Thank you to the reviewer for their insightful remarks. Several images (along with dimensions) have been added and the text edited to read more smoothly in American English. Dimensions for all works are also included, as suggested. Additional text has been added to clarify the patriotic importance of both apple trees and old manor houses. Tense irregularities have been corrected and the intro and abstract expanded and a new conclusion added, since the original conclusion has now been relocated to the intro section. The bracket irregularities have also been corrected.